# Transportation Infrastructure Construction and High-Quality Development of Enterprises: Evidence from the Quasi-Natural Experiment of High-Speed Railway Opening in China

**Tianjiao Zhao [1], Xiang Xiao [2,3,\*] and Qinghui Dai [2]**

1    School of Economics and Management, University of Science and Technology Beijing, Beijing 100083, China; zhaotj@ustb.edu.cn
2    School of Economics and Management, Beijing Jiaotong University, Beijing 100044, China; 18113069@bjtu.edu.cn
3    Central and Eastern Europe Research Centre, Beijing Jiaotong University, Beijing 100044, China
\*    Correspondence: xxiao@bjtu.edu.cn

**Abstract:** High-quality development of the economy is an important guarantee for economic and business sustainability, and the construction of transportation infrastructure is an important channel to achieving high-quality development. Thus, we take the opening of China's high-speed railway (*HSR*) as a quasi-natural experiment and use the difference-in-difference model to explore the impact and mechanism of *HSR* on firms' high-quality development. By using the total factor productivity of enterprises as the proxy for high-quality development, the empirical results show that: (1) the opening of the *HSR* can significantly promote high-quality development of enterprises; (2) the quality of information disclosure plays a mediating role in such relationships; and (3) the impact of *HSR* on enterprises' high-quality development is more significant for enterprises that are located in cities with better business environments. Overall, this research indicates that local infrastructure construction is an important factor to achieve high-quality development of enterprises as well as economic sustainability that cannot be ignored, and this conclusion will be helpful for corporate managers in enhancing the quality of information disclosure, as well as for local governments to attach more importance to optimizing business environments to achieve high-quality development and economic sustainability.

**Keywords:** transportation infrastructure construction; high-speed railway; high-quality development; information disclosure quality; business environment

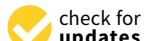



## 1. Introduction

China's economy has shifted from a stage of high-speed growth to a stage of high-quality development in recent years. The stage of "high-quality development" is a new development concept, featuring innovative, coordinated, green, open, and shared development, which provides an important guarantee for the sustainable development of the economy. Investment in transportation infrastructure has long been considered one of the key factors in promoting economic growth [1], and it is also an important guarantee and driving force for accelerating regional development [2]. As the main body of the economy and society, enterprises must be good at grasping the general trend and summing up experiences in order to meet the requirements of national macroeconomic development policies and comply with the development of the new era. Only by formulating and perfecting the road map of high-quality development, constantly improving the operation quality, and enhancing their competitiveness can enterprises realize high-quality and sustainable development.

High-Speed Railway (*HSR*) is a large-scale transportation infrastructure investment that was introduced in China in 2008 to facilitate the flow of information, capital, and labor

among cities [3,4], to increase spatial and social equity [5–7], and to stimulate economic growth [8,9]. Compared with traditional means of transportation, *HSR* has a significant space–time compression effect and a profound impact on social and economic operations. Since 1 August 2008, China's first Beijing–Tianjin intercity railway with fully independent intellectual property rights and a speed of up to 350 km/h has been in operation, marking China's official step into the era of *HSR*. Since then, China has quickly embarked on large-scale *HSR* development. By the end of 2020, the total length of the *HSR* in operation in China had reached 38,000 km, and the proportion of *HSR* operating mileage of the overall railway operating mileage also showed a rapid rise, from 10.68% in 2013 to 25.97% in 2020. China has become a country with the longest *HSR* mileage, the highest transportation density, and the most complex network operation in the world [10]. *HSR* is becoming an important way for people to travel because of its speed, convenience, and punctuality. Compared with other traditional modes of transportation, *HSR* has unique advantages in terms of shortening the travel time and space and realizing large-scale personnel transfers. *HSR* has made it easier for people to travel from station to station and has had a huge impact on every aspect of economic life.

Previous studies have provided mixed results on the impact of *HSR* on the local economy. Some studies have shown that the opening of the *HSR* can promote the process of economic integration, spread the resources of the central city to the surrounding cities, and promote the economic growth of the surrounding cities [11]. Ahlfeldt and Feddersen found that the construction of the *HSR* greatly optimized the original transportation network, reduced the space and time distances, and had a positive impact on regional economic growth. This includes promoting the formation of urban agglomeration along the railway, promoting regional economic integration, saving social costs, increasing the employment level, improving market access environment, and increasing the value of agricultural land [12]. At the same time, the opening of the *HSR* also speeds up the flow of knowledge and information between different regions [13], reducing the cost of information acquisition and communication. However, while the economic diffusion effect of the *HSR* is supported, some scholars have also proposed that the opening of the *HSR* will accelerate the transfer of factor resources from surrounding cities to central cities, inhibit the economic growth of adjacent non-central cities, and eventually lead to the widening of the regional economic gap [8]. Thus, the economic and social effects of the *HSR* are double-sided.

Although transportation infrastructure may reshape business activities, leading to enhanced economic growth [14], there is little is known about the impact of transportation infrastructure construction on micro enterprises and their sustainable development. Most of the related research has been focused on the influence of geographical distance on micro enterprises' payout policy [15], venture capital investment [16], audit quality [17], analyst coverage [18], and the accuracy of analyst's forecasts [19], while *HSR* is represented as an effective way to reduce spatial and geographical distance [1,10], and the spatiotemporal squeezing effect generated by *HSR* has been shown to significantly improves the accessibility of the local firms [20]. Prior literature about the opening of *HSR* has shown that it will significantly improve enterprises' *CSR* performance [21,22], and increase the value of tourism firms [23]. In addition, Li and Chan found that the number of site visits by analysts and the number of analysts involved in the site visits increased significantly after the opening of the *HSR* in the cities where the listed firms were located [20]. Zheng et al. found that the increased investor site visits caused by the *HSR* openings were the internal mechanism by which *HSR* improved companies' *CSR* performances [22]. Therefore, there have been few studies on how *HSR* affects the high-quality development of enterprises.

Based on information asymmetry theory and resource dependence theory, this paper takes the opening of *HSR* in China as a quasi-natural experiment and explores the impact of transportation infrastructure construction on the high-quality development of enterprises. One of the main advantages brought about by the opening of *HSR* to local enterprises is the improved efficiency of information exchange with the external capital market, which increases the degree of external supervision of enterprises. Meanwhile, enterprises also

hope to expand their business networks through the opening of *HSR*, so the quality of the information disclosure of enterprises must be improved accordingly. Thus, this research explores the specific *HSR* mechanism affecting the high-quality development of enterprises from the perspective of enterprise information disclosure quality. Moreover, the regional business environment will be affected by the effect of the *HSR* on enterprise development. Enterprises in regions with a better business environment have greater government support for their development and are more likely to take advantage of the convenience of *HSR* to create value. Thus, in this study, we further investigated the moderating role of the regional business environment on the relationship between *HSR* and enterprises' high-quality development.

Taking the Chinese A-share listed companies that publicly traded in the Shanghai Stock Exchange (*SSE*) and Shenzhen Stock Exchange (*SZSE*) from 2003 to 2019 as the research sample, we adopted a difference-in-difference (*DID*) approach to empirically examine the influence of the *HSR* opening on firms' high-quality development, as well as the specific influencing mechanism from the perspective of information disclosure quality. The results showed that: first, the total factor productivity of Chinese A-share listed companies was significantly improved after the companies experienced *HSR* openings. Second, information disclosure quality played a mediating role in explaining the positive effect of the *HSR* opening on enterprises' total factor productivity. The stepwise regressions showed that the opening of *HSR* significantly improved the quality of information disclosure of enterprises, which is reflected in the reduction of enterprises' earnings management, and the opening of *HSR* and the higher quality of information disclosure jointly promoted the improvement of the total factor productivity of enterprises. The results remained stable under the placebo test and when using alternative measurements for the high-quality development of enterprises. To mitigate the reverse causality at the firm level, we adopted the following robustness checks: (1) excluding railway pivot cities (municipalities and provincial capitals); (2) employing the propensity score matching method; (3) taking the slope of the prefecture-level city as the instrumental variable of whether a city has *HSR*. All three methods supported the baseline findings. Further analysis showed that the regional business environment moderated the impact of the opening of *HSR* on the high-quality development of enterprises. The positive relationship became more pronounced for enterprises that were located in cities with better business environments.

This paper contributes to the existing literature in the following ways: first, using *HSR* openings as an exogenous shock, this research explores the impact of transportation infrastructure construction on enterprises' high-quality development, both theoretically and empirically. Current *HSR*-related studies have mostly focused on the macroeconomic consequences of *HSR* and its impact on urban development, such as the economic growth [1], development of urban tourism [24], and the quality of urban economic growth [25]. Limited attention has been paid to the micro enterprise level [9], especially in terms of the high-quality development of enterprises. Based on information asymmetry theory and stakeholder theory, this paper carries out a theoretical analysis of how the opening of *HSR* affects the high-quality development of enterprises, expanding the micro level of the existing research on the economic consequences of *HSR*. Second, previous studies have mostly discussed the mechanisms of *HSR's* micro-effect from the perspective of public pressure [21] and site visits by analysts or investors [20,22,26], while this paper examines the specific mechanism of *HSR* opening in promoting the listed enterprises' high-quality development from the perspective of information disclosure quality, offering empirical evidence of the improvement of firm-specific information transparency caused by the *HSR* opening. Third, combined with the geographical characteristics of China's vast territory as well as the fact that local governments provide different levels of support for enterprise development, this research further explores how the opening of *HSR* affects the high-quality development of enterprises by taking into account the differences in regional business environments, which provides theoretical support and practical inspiration for optimizing the regional business environment in China and achieving the goal of high-quality development.



The rest of the paper is organized as follows. We start with a description of the institutional background in Section 2. Section 3 contains the literature review and hypotheses development. Section 4 provides the methodology, including the data and sample, research models, and the measurement of variables. Section 5 presents the empirical results and analysis, Section 6 is the discussion and Section 7 offers concluding remarks.

## 2. Institutional Background

Fu et al. concluded that *HSR* has had a more significant impact on China's economy than on other countries [27]. This is mainly because *HSR* plays a more significant economic and social role in China.

First of all, there is a saying in China that "If you want to get rich, build roads first". It means that roads, as an infrastructure project, are of self-evident importance. Because of China's vast territory, China is currently facing the problem of unbalanced regional economic development. To achieve economic and business sustainability, the train has become the main form of transportation which supports large and frequent flows of talent and materials. In the past, the "green train" has become the mainstream of railway passenger transport. However, owing to the characteristics of the traditional train, such as the low speed, multiple stops, short distances between stations, and low transportation efficiency, there is a serious shortage in terms of railway passenger and freight transportation capacity. The opening of *HSR* has significantly alleviated these problems.

The economic effect of *HSR* is mainly observed in promoting the coordinated development of the regional economy [13,21]. The coordinated development of the regional economy is an important strategy in China's development and is related to whether China can achieve the ultimate goal of common prosperity. In the central and western regions of China, owing to the lack of transportation facilities, personnel exchanges and material flows are blocked, which makes it difficult for advanced technologies to be widely and quickly promoted. In addition, because of the geographical constraints, there is no advantage to investing funds. Therefore, many companies will prioritize sending resources to the east, where education levels and research capabilities are stronger. This creates a vicious circle of widening regional disparities. The opening of *HSR* could narrow the economic gap between developed and less developed regions. Northeast China, as an old industrial base, has suffered from a backward GDP and serious population outflow in recent years. The opening of Beijing–Shenyang and Harbin–Dalian *HSR*s has significantly alleviated the economic weakness of the three provinces in northeast China.

Secondly, *HSR* plays a significant social role in China, helping to promote the sustainable development of China's economy [25]. In China, the social effects of *HSR* are mainly reflected in the following three aspects. (1) *HSR* improves the quality of residents' travel. On the one hand, residents' travel behavior and choices will change because of the improvement of transportation facilities brought about by *HSR* [27]; on the other hand, the improvement of regional accessibility enhances travel efficiency, and thus improves residents' travel quality [28]. (2) It accelerates the inflow of talent and promotes employment. In the era of the knowledge economy, human capital is not only the basis for improving enterprise performance and promoting enterprise innovation, but is also the key factor for maintaining long-term economic growth. The advantages of high speed and punctuality can promote the flow of human capital with high time value and attract higher-level talents to work in cities with *HSR*s [13]. (3) It improves the industrial structure. On the one hand, the construction of *HSR* can drive the development of a large number of enterprises, such as iron and steel, machinery, materials, and other industries. On the other hand, *HSR* promotes the development of electronics, computer, communication, and other industries, and forms new high-tech industrial clusters. Correspondingly, less developed areas can take on the industrial transfer of the developed areas and solve the problems of their lack of linkage with surrounding areas and the transportation shortage. Therefore, *HSR* has displayed a wide range of social effects.

In addition, China has been vigorously reforming its business environment in recent years. According to the Doing Business 2020 report released by the World Bank, China ranks 31st in the world, with an upward trend year by year. The Regulation on Optimizing the Business Environment, which took effect in 2020, provided institutional guarantees for improving China's business environment. However, owing to China's vast territory, the business environments of China's 31 provinces vary greatly, and local governments also show great differences in their support for enterprise development. Therefore, the economic and social effects of the opening of the *HSR* may have different impacts on the high-quality development of enterprises.

Thus, because of China's particular situation, *HSR* has a significant social and economic effect; thus, this paper discusses whether the opening of *HSR* contributes to the realization of the high-quality development strategy of enterprises, and such a relationship is affected by the regional business environment.

## 3. Literature Review and Hypotheses Development

### 3.1. HSR and High-Quality Development of Enterprises

As a new means of transportation, *HSR* offers the benefits of high speed, environmental protection, punctuality, and safety [28], and it can shorten the time and space between economic entities. Although information and network technologies are very developed in today's society, the geographical location of an enterprise still plays an important role in its transaction mode, investment and financing behavior, operating profit, and capital market transaction structure [29]. Prior literature has shown that geographical distance is the key factor hindering information acquisition [30]. Based on the theory of information asymmetry, geographical distance will affect economic subjects through two channels. On the one hand, geographical distance affects the information acquisition cost to economic subjects. When the geographical distance is great, the cost to economic subjects of obtaining external information becomes extremely high, and the regional economic development also faces serious obstacles. In addition, Coval and Moskowitz found that local fund managers can earn higher returns by investing in local stocks [31]. Malloy found that sell-side analysts have better earnings forecasts for geographically close companies [19]. Proximity also helps banks to gather private information about small businesses, making it easier for them to get bank loans [32]. On the other hand, geographical distance affects the degree of external supervision of the economic subject [10]. Greater geographical distance hinders external supervisors from effectively supervising economic subjects and promotes the "agency problem". Numerous studies have supported this idea, such as that of Gaspar and Massa, which found that local shareholders are more effective in improving governance [33]; Opie et al. found that when local state-owned enterprises operate far away from the controlling shareholder's location, the controlling shareholder has difficulty in effectively supervising the enterprise, resulting in the poor investment efficiency of local state-owned enterprises [34]. Proximity helps regulators to probe companies [35]. The opening of *HSR* can accelerate the flow of personnel and capital, speeding up the dissemination of information in different regions, thus reducing the cost of information acquisition and communication, and providing convenient supervision conditions for external supervisors, thus improving the information transparency of the enterprise.

On the one hand, the opening of *HSR* can effectively reduce the information acquisition cost to outsiders, enhance the transparency of corporate information, and effectively improve the corporate governance environment. Private information is often more valuable than public information, and prior research has found that local investors can get extra returns by investing in local enterprises because of their information advantage [36]. Corporate site visits are an important way for market intermediaries to obtain first hand non-public information [26]. Hauswald and Marquez found that the advantages of geographical proximity help analysts, banks, and venture capitalists to collect and analyze private information, so as to obtain more accurate information [37]. Cheng et al. also found that corporate site visits significantly affect stock returns [38]. The opening of *HSR* can

effectively alleviate the problem of the high degree of information opacity caused by the lack of institutional investor research and analysts' inattention to enterprises in remote areas. Some studies have found that the *HSR* opening is beneficial for investors in obtaining soft information [39], increasing the number of site visits by analysts [20], and is helpful for improving the accuracy of analysts' earnings forecasts [26]. Therefore, the increased information transparency of listed companies brought about by the opening of the *HSR* is conducive to the high-quality development of enterprises.

On the other hand, the opening of *HSR* provides a convenient channel for outsiders to monitor corporate behavior. The opening of *HSR* can break the geographical separation between listed companies and stakeholders, and greatly stimulate the enthusiasm of independent directors and auditors. Prior research showed that frequent visits by the audit engagement partner and senior manager to an audit site are one of the highest-rated attributes of audit quality; thus, the opening of *HSR* can improve audit quality [17]. The active performance of governance responsibilities can help reduce information asymmetry between economic subjects and increase the discovery of hidden negative news of listed companies. *HSR*, with its advantages of high speed, high passenger capacity, and high punctuality rates, can effectively shorten people's travel time and reduce people's travel costs [28], which is conducive to increasing the number of site visits by certified public accountants to the enterprise, and makes it convenient for auditors and regulators to supervise enterprises [40], thus reducing the possibility of enterprises hiding negative news.

In addition, the opening of *HSR* helps to increase the media attention to listed enterprises. Once negative news about listed companies is exposed, the related media reports will significantly increase, compared with those companies located in cities without *HSR*, and the negative media reports will have a negative impact on the reputations of listed companies. Therefore, the enhanced external supervision brought about by the opening of *HSR* can help to promote the high-quality development of enterprises and avoid negative events.

Thus, we propose the following testable hypothesis:

**Hypothesis 1 (H1).** *The opening of HSR will help promote high-quality development of enterprises.*

### 3.2. The Mediating Role of Information Disclosure Quality

Geographical location will affect the economic decision making and governance level of listed companies. We argue that the opening of *HSR* may affect the high-quality development of enterprises by way of affecting the quality of corporate information disclosure.

First of all, the opening of *HSR* helps improve enterprises' information disclosure quality. On the one hand, based on the information asymmetry theory and agency theory, previous studies have found that a long geographical distance will cause information asymmetry and thus lead to more agency conflicts [41]. However, the opening of *HSR* has increased the number of site visits by institutional investors and auditors, which is convenient for external supervisors, thus minimizing the agency problem between the shareholders and management, as well as the possibility of earnings manipulation [20], thus improving the quality of the information disclosure of enterprises. The enterprise is under external supervision and under pressure to passively improve the quality of information disclosure. Such external supervision and pressure urge enterprises to passively improve the quality of information disclosure.

On the other hand, based on the resource dependence theory, since enterprises are not independently operated and developed, the sustainable development of enterprises needs to rely on external organizational resources. The opening of *HSR* provides convenient conditions for local enterprises to exchange information and other resources with the outside world. Therefore, enterprises hope to attract external investors or partners through a higher quality of information disclosure and expand their business with more extensive resources. Thus, the opening of *HSR* is one of the internal motivations for enterprises to improve the quality of information disclosure.

Secondly, higher quality of information disclosure is conducive to the high-quality development of enterprises. The higher quality of enterprise information disclosure helps them gain favor from external stakeholders. On the one hand, enterprises with better information disclosure quality are more likely to attract external investors, enabling them to more easily obtain financing, and thus improving their investment efficiency [42]. On the other hand, better information disclosure is also conducive to promoting the business relationship between enterprises and their upstream and downstream partners in the industrial chain, improving the total factor productivity of enterprises through substantive business, and achieving the goal of high-quality development.

Thus, we propose Hypothesis 2:

**Hypothesis 2 (H2).** *Information disclosure quality mediates the impact of HSR opening on firms' high-quality development.*

### 4. Methodology

*4.1. Data and Sample*

Our research sample contained all the A-share listed firms that publicly traded in the SSE and SZSE during 2003–2019. The sample period began in 2003 because some variables in the regression model were not available prior to 2003. All of the financial data were extracted from the China Stock Market and Accounting Research (CSMAR) database. To enhance measurement validity, we eliminated the initial sample as follows: (1) firms operating in the financial sector; (2) financially distressed firms; (3) samples with incomplete data. Finally, we obtained 26,245 unbalanced firm-year observations. All of the continuous variables were winsorized at the 1% and 99% levels.

*4.2. Variables*

4.2.1. High-Quality Development of Enterprises

High-quality development of enterprises refers to the pursuit of a high level and high efficiency of economic value and social value creation, with innovation serving as the first impetus [43]. According to this definition, the total factor productivity (*TFP*) of enterprises was a suitable proxy for high-quality development, since Robert Merton Solow, a Nobel Prize winner of economics, attributed the *TFP* to technological progress. Non-parametric methods for estimating the *TFP* were mostly improved based on the methods proposed by Levinsohn and Petrin [44]. Levinsohn and Petrin used intermediate inputs rather than investment as the proxy to avoid the estimation bias caused by enterprises with missing investment values or negative values [44,45]. Thus, we adopted the method proposed by Levinsohn and Petrin to estimate the *TFP* in our research [44]. Following Levinsohn and Petrin, we calculated enterprises' total factor productivity (*TFP*) through the following model (1).

$$\ln(Y_{i,t}) = \beta_0 + \beta_l \ln(L_{i,t}) + \beta_k \ln(K_{i,t}) + \beta_m \ln(M_{i,t}) + \omega_{i,t} \tag{1}$$

where *Y* is the logarithm of the enterprises' output, which is measured by operating revenue; *L* represents the labor input, which is measured by the number of employees; *K* represents the capital input, which is measured by the net fixed assets of enterprises, and *M* is the intermediate goods input, which is measured by the actual cash paid by the enterprise to purchase goods and receive services. The output variable, intermediate input variable, and capital input variable were deflated by the industrial producer's ex-factory price index, the industrial producer's purchase price index, and the fixed asset investment price index, respectively. Taking the natural logarithm of the residual value, we obtain the level of the *TFP* of enterprises under the LP method.

4.2.2. Information Disclosure Quality

Referring to the prior literature, we used two measures of earnings management as the proxy for the information disclosure quality of the listed enterprises, namely, accrual earnings management and real activity earnings management. Following Yao and Liu [46],

the accrual earnings management (*EM_Accrual*) was calculated through the modified Jones model that is shown in Equation (2) [47].

$$\frac{Accruals_t}{A_{t-1}} = \alpha_0 + \alpha_1 \frac{1}{A_{t-1}} + \alpha_2 \frac{\Delta S_t - \Delta REC_t}{A_{t-1}} + \alpha_3 \frac{PPE_t}{A_{t-1}} + \varepsilon_t \tag{2}$$

where *Accruals* represents the total accrual earnings, *A* represents the total assets, *ΔS* represents the change in sales, *ΔREC* represents the change in accounts receivable, and *PPE* represents net fixed assets. The residual *ε* represents a firm's accrual earnings management. A higher value of *EM_Accrual* means that the information disclosure quality of the enterprise is lower.

Following Dechow et al. and Roychowdhury [48,49], the real activity earnings management (*EM_Real*) was calculated through the following equations:

$$\frac{CFO_t}{A_{t-1}} = \beta_0 + \beta_1 \frac{1}{A_{t-1}} + \beta_2 \frac{REV_t}{A_{t-1}} + \beta_3 \frac{\Delta REV_t}{A_{t-1}} + \mu_t \tag{3}$$

$$\frac{PROD_t}{A_{t-1}} = \gamma_0 + \gamma_1 \frac{1}{A_{t-1}} + \gamma_2 \frac{REV_t}{A_{t-1}} + \gamma_3 \frac{\Delta REV_t}{A_{t-1}} + \gamma_4 \frac{\Delta REV_{t-1}}{A_{t-1}} + u_t \tag{4}$$

$$\frac{DISEXP_t}{A_{t-1}} = \delta_0 + \delta_1 \frac{1}{A_{t-1}} + \delta_2 \frac{REV_{t-1}}{A_{t-1}} + v_t \tag{5}$$

where *CFO* represents the net cash flow from operations; *PROD* represents the cost of production of an enterprise; *DISEXP* is the operating expenses of an enterprise, which is equal to the sum of the selling expenses and administrative expenses of the enterprise. *A* represents the total assets, *REV* represents the revenue of the enterprise, *ΔREV* represents the change in revenue, and *PPE* represents net fixed assets. The residual *μ*, *u*, and *v* represent abnormal net cash flow from operations, abnormal cost of production, and abnormal operating expenses, respectively. The degree of real activity earnings management (*EM_Real*) was calculated as the abnormal cost of production minus the abnormal net cash flow from operations, and minus the abnormal operating expenses. A higher value of *EM_Real* means that the information disclosure quality of the enterprise is lower.

*4.3. Research Methods*

4.3.1. Baseline Model

We adopted the *DID* approach to estimate the impact of the *HSR* opening on the *TFP* among the Chinese listed enterprises. Since the opening times of the *HSR* in various cities were different, we employed the staggered *DID* approach with year fixed effect and industry fixed effect, following prior research [9,50].

$$TFP_{i,t} = \rho_0 + \rho_1 HSR_{i,t} + \sum_j \rho_j Controls_{i,t} + \sum Ind + \sum Year + \varepsilon_{i,t} \tag{6}$$

where $TFP_{i,t}$ is the total factor productivity of the firm *i* in year *t*, $HSR_{i,t}$ is the dummy variable indicating whether the city where firm *i* is located opened an *HSR* in year *t*. Our primary interest is the coefficient $\rho_1$ because it captures the impact of the *HSR* opening on *TFP*. The control variables in Equation (6) include firm size (*SIZE*), financial leverage (*LEV*), firm performance (*ROE*), growth rate of total asset (*GROWTH*), book to market value (*BtoM*), cash flow of the enterprise (*Cashflow*), state ownership (*STATE*), the difference between control right and ownership right (*DUAL*), shareholding ratio of the largest shareholder (*Top1*), and the growth rate of the GDP (*GDPgrt*). The variable definitions are shown in Appendix A.

4.3.2. The Mediating Role of Information Disclosure Quality

To test H2, we adopted the stepwise regression method proposed by Baron and Kenny and used the following models to verify the mediation effect of information disclosure quality [51].

$$AM\_Accrual_{i,t} = \eta_0 + \eta_1 HSR_{i,t} + \sum_j \eta_j Controls_{i,t} + \varepsilon_{i,t} \tag{7}$$

$$AM\_Real_{i,t} = \theta_0 + \theta_1 HSR_{i,t} + \sum_j \theta_j Controls_{i,t} + \varepsilon_{i,t} \tag{8}$$

$$TFP_{i,t} = \lambda_0 + \lambda_1 HSR_{i,t} + \lambda_2 AM\_Accrual_{i,t} + \sum_j \lambda_j Controls_{i,t} + \varepsilon_{i,t} \tag{9}$$

$$TFP_{i,t} = \phi_0 + \phi_1 HSR_{i,t} + \phi_2 AM\_Real_{i,t} + \sum_j \phi_j Controls_{i,t} + \varepsilon_{i,t} \tag{10}$$

Models (6), (7) and (9) were used to jointly test the mediating effect of accrual earnings management on the path of the *HSR* opening affecting enterprises' total factor productivity, while models (6), (8) and (10) were used to test the mediating effect of real activity earnings management. The mediating effect of information disclosure quality was calculated by STATA, including the results of the Sobel test, Goodman test 1, and Goodman test 2.

## 5. Results and Discussion

*5.1. Summary Statistics*

Table 1 presents the descriptive statistics of the full sample. The first two columns show the mean value and standard deviation of each variable, and the last three columns are the 25% value, median value, and 75% value, respectively. The definitions of the variables are presented in Appendix A.

**Table 1.** Summary statistics.

| Variable | Mean | Std. | 25% | Median | 75% |
|---|---|---|---|---|---|
| *TFP* | 8.0693 | 1.0107 | 7.3718 | 7.9937 | 8.6751 |
| *HSR* | 0.5369 | 0.4986 | 0 | 1 | 1 |
| *SIZE* | 22.0823 | 1.2655 | 21.1786 | 21.9083 | 22.7974 |
| *LEV* | 0.4593 | 0.2003 | 0.3063 | 0.4631 | 0.6124 |
| *ROE* | 0.0493 | 0.1552 | 0.0247 | 0.0641 | 0.1113 |
| *Growth* | 0.1703 | 0.3354 | 0.0098 | 0.0940 | 0.2210 |
| *BtoM* | 0.6507 | 0.2489 | 0.4572 | 0.6624 | 0.8549 |
| *Cashflow* | −1.5407 | 4.1458 | −1.0495 | −0.4042 | −0.1787 |
| *STATE* | 0.4810 | 0.4996 | 0 | 0 | 1 |
| *DUAL* | 5.0777 | 7.8133 | 0 | 0 | 9.1299 |
| *Top1* | 35.8807 | 15.2457 | 23.7800 | 33.6600 | 46.6300 |
| *EM_Accrual* | 0.0118 | 0.0995 | −0.0392 | 0.0112 | 0.0611 |
| *EM_Real* | 0.0002 | 0.1974 | −0.0929 | 0.0097 | 0.1043 |
| *GDPgrt* | 0.1171 | 0.0510 | 0.0833 | 0.1027 | 0.1483 |

The results of Table 1 revealed that the mean *TFP* was 8.07, and the standard deviation was 1.01. In our sample, 54% of the observations had experienced the opening of *HSR*. The standard deviations of the control variables *Cashflow*, *DUAL*, and *Top1* were relatively large, which indicated that the cash flow and the corporate governance of each sample firm were quite different.

*5.2. Baseline Results*

5.2.1. *HSR* and *TFP*

Table 2 presents the baseline results of the difference-in-difference regression. It shows the effect of *HSR* on the firms' total factor productivity. Column (1) presents the regression result that was not controlled for the year fixed effect or industry fixed effect,

while column (2) presents the result that was controlled for both. The results revealed that the coefficients of *HSR* in both columns were positive and significant at the 1% level, indicating that the *TFP* of enterprises in cities with *HSR* is significantly improved after the *HSR* opening, thus the *HSR* opening promotes the high-quality development of enterprises. Additionally, the results of the control variables showed that firm size (*SIZE*), financial leverage (*LEV*), firm performance (*ROE*), cash flow of the enterprise (*Cashflow*), corporate governance (*DUAL*), and the growth rate of GDP (*GDPgrt*) were all positively related to the *TFP*. These results were consistent with the prior literature. Therefore, our hypothesis 1 was verified.

**Table 2.** Baseline regression results.

|  | (1) | (2) |
|---|---|---|
|  | *TFP* | *TFP* |
| *HSR* | 0.0276 *** | 0.0214 *** |
|  | (3.5223) | (2.9283) |
| *SIZE* | 0.6138 *** | 0.6351 *** |
|  | (128.5846) | (129.3674) |
| *LEV* | 0.7733 *** | 0.6611 *** |
|  | (32.6781) | (27.8619) |
| *ROE* | 0.9779 *** | 0.9327 *** |
|  | (27.3383) | (27.3119) |
| *Growth* | −0.1446 *** | −0.1175 *** |
|  | (−10.9278) | (−9.5345) |
| *BtoM* | −0.1126 *** | −0.1931 *** |
|  | (−6.3459) | (−9.0936) |
| *Cashflow* | 0.0120 *** | 0.0093 *** |
|  | (11.7626) | (9.5945) |
| *STATE* | −0.0304 *** | 0.0173 ** |
|  | (−3.5952) | (2.1599) |
| *DUAL* | 0.0026 *** | 0.0009 ** |
|  | (5.2446) | (2.0964) |
| *Top1* | 0.0017 *** | 0.0023 *** |
|  | (6.2637) | (9.6472) |
| *GDPgrt* | 0.7348 *** | 0.2207 * |
|  | (8.7863) | (1.6953) |
| *INDUSTRY FE* | NO | YES |
| *YEAR FE* | NO | YES |
| Intercept | −5.9160 *** | −6.4900 *** |
|  | (−59.1043) | (−63.7267) |
| *N* | 26,245 | 26,245 |
| Adj $R^2$ | 0.6579 | 0.7326 |
| *F* | 4827.65 | 1205.62 |

Note: t-statistics in parentheses. *** $p < 0.01$, ** $p < 0.05$, and * $p < 0.1$.

The preconditions for difference-in-difference were the parallel trend between the *HSR*-affected firms in the treatment group and the unaffected firms in the control group. Thus, we performed a parallel trend test. In the parallel trend test, we introduced eight dummies: *pre_3*, *pre_2*, *pre_1*, *current*, *post_1*, *post_2*, *post_3* and *post_4*. The dummy *current* was set to one if it was the year of the *HSR* opening for the city where an enterprise was located; *pre_3*, *pre_2*, and *pre_1* were the three pretreatment years, and *post_1*, *post_2*, *post_3* and *post_4* were the four post-treatment years. Figure 1 shows the parallel trend test result.

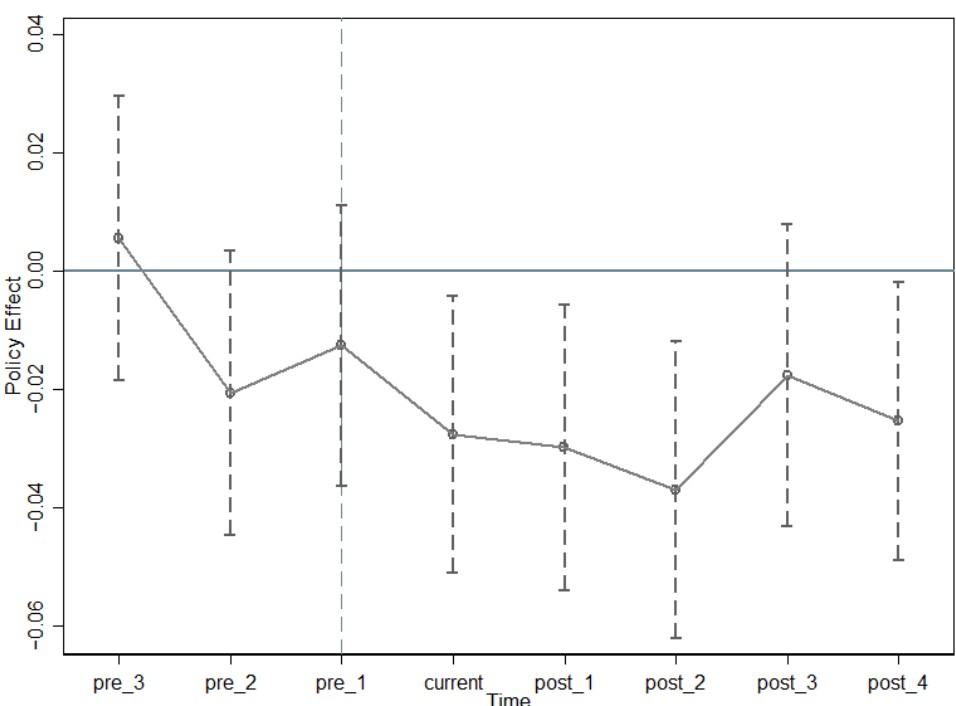

**Figure 1.** Parallel trend test.

It can be seen from Figure 1 that the coefficient of the interaction term was not significantly different from 0 before the policy (90% confidence interval contains 0 value), indicating that there was no significant difference between the treatment group and the control group before the time of the policy. Therefore, it met the requirements of the parallel trend of the *DID* test. In the current and future policy period, the interaction coefficient was significantly different from 0 at a 90% confidence level, indicating that the opening of *HSR* indeed had a significant impact.

A potential concern with our identification stemmed from the confounding factors driving the high-quality development of the listed enterprises, such as the level of economic development and future economic growth of a city. To disentangle the actual *HSR* effect from these confounding factors, we conducted a placebo test by randomly assigning an opening year to each city that had an *HSR* opening in our sample period. The results are shown in Figure 2.

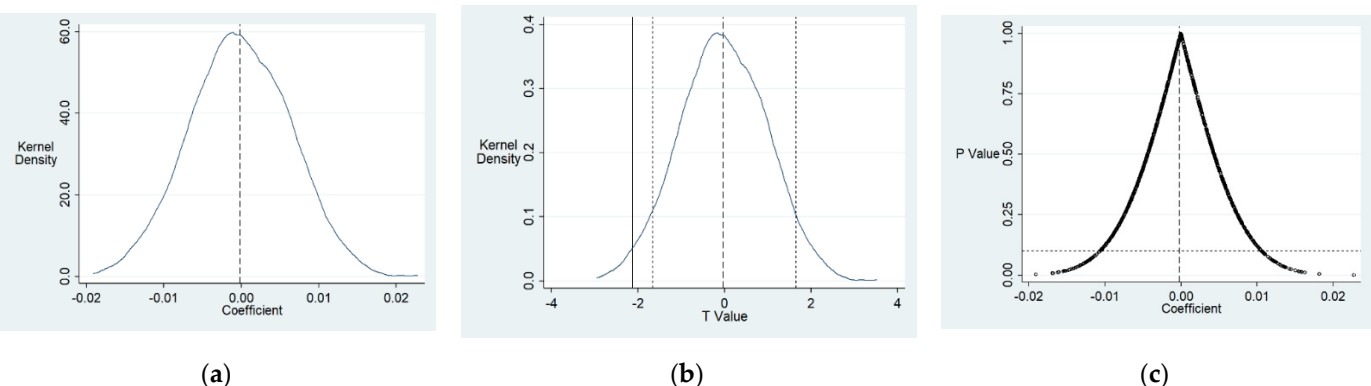

|(**a**)|(**b**)|(**c**)|

**Figure 2.** Placebo test (**a**) Kernel density of coefficients; (**b**) Kernel density of T value and (**c**) Scatter diagram of P values.

Figure 2a is the kernel density estimation of the regression coefficients of *HSR* after randomization. Figure 2b is the kernel density estimation of the T value, where the solid

line is the true T value, and the dotted line is the "virtual" mean of 1000 regressions. The two short, dashed lines in Figure 2b represent T equal to −1.65 and 1.65, that is, the T value corresponding to the significance level of 10%. The T value of the placebo test was in the range of −1.65 to 1.65, indicating that it was not significant, at least at the 10% level. Figure 2a,b both illustrate a basic fact that most of the coefficients and T values were concentrated around 0, the mean value of coefficients and T values was far away from the true value, and most of the estimated coefficients were insignificant. Figure 2c is the scatter diagram of *p* values, where the horizontal short, dashed line represents a *p* value equal to 0.1. Most of the scattered points in Figure 2c are above the dotted line, indicating that they were insignificant at the 10% level. Thus, the placebo test results revealed that the impact of the *HSR* opening on the high-quality development of enterprises was not affected by other unobserved factors.

### 5.2.2. The Mediating Role of Information Disclosure Quality

Table 3 presents the empirical results for how information disclosure quality mediated the impact of *HSR* on the high-quality development of enterprises. The first three columns show the mediating effect of information disclosure quality that are proxied by accrual earnings management (*EM_Accrual*), while the last three columns show the mediating effect of information disclosure quality that are proxied by real activity earnings management (*EM_Real*). The coefficient of *HSR* in column (1) represents that *HSR* opening has a significant positive influence on the *TFP*; column (2) shows that the opening of *HSR* significantly inhibit firms' earnings management, which leads to a better information disclosure quality; and column (3) presents the joint effect of *HSR* and information disclosure quality on the *TFP,* indicating that *HSR* and information quality significantly enhanced the *TFP* of the listed enterprises jointly. According to Baron and Kenney [51], the core explanatory variables in the stepwise regressions were all significant, suggesting that information disclosure quality played a mediating role in the way that *HSR* facilitates firms' *TFP*, and the mediating effect was approximately 7.04%. The last three columns of Table 3 show the results of the mediating effect of information disclosure quality that were proxied by real activity earnings management. The results in columns (4) to (6) are similar to those in columns (1) to (3), with a mediating effect of 5.15%. Additionally, the Sobel test, Goodman test 1, and Goodman test 2 on the two groups of successive regressions were statistically significant. Therefore, we can conclude that information disclosure quality does play a partial mediating role in the channels through which *HSR* improves enterprises' development quality, which supports our H2.

**Table 3.** The mediating role of information disclosure quality.

| | (1) TFP | (2) EM_Accrual | (3) TFP | (4) TFP | (5) EM_Real | (6) TFP |
|---|---|---|---|---|---|---|
| HSR | 0.0214 *** | −0.0042 *** | 0.0199 *** | 0.0214 *** | −0.0059 ** | 0.0203 *** |
| | (2.9283) | (−3.3569) | (2.7287) | (2.9283) | (−2.2373) | (2.7870) |
| EM_Accrual | | | −0.3638 *** | | | |
| | | | (−8.8824) | | | |
| EM_Real | | | | | | −0.1862 *** |
| | | | | | | (−9.6910) |
| SIZE | 0.6351 *** | 0.0040 *** | 0.6365 *** | 0.6351 *** | −0.0233 *** | 0.6307 *** |
| | (129.3674) | (4.8196) | (129.9852) | (129.3674) | (−12.6853) | (128.6847) |
| LEV | 0.6611 *** | −0.0351 *** | 0.6483 *** | 0.6611 *** | 0.1410 *** | 0.6873 *** |
| | (27.8619) | (−9.2629) | (27.3622) | (27.8619) | (17.9775) | (28.8338) |
| ROE | 0.9327 *** | 0.1670 *** | 0.9934 *** | 0.9327 *** | −0.2061 *** | 0.8943 *** |
| | (27.3119) | (31.9889) | (28.3383) | (27.3119) | (−20.8581) | (26.1328) |
| Growth | −0.1175 *** | 0.0344 *** | −0.1050 *** | −0.1175 *** | 0.0384 *** | −0.1104 *** |
| | (−9.5345) | (12.0503) | (−8.4621) | (−9.5345) | (6.1748) | (−8.9154) |
| BtoM | −0.1931 *** | 0.0074 ** | −0.1904 *** | −0.1931 *** | 0.2108 *** | −0.1538 *** |
| | (−9.0936) | (2.1148) | (−8.9747) | (−9.0936) | (26.7579) | (−7.1008) |
| Cashflow | 0.0093 *** | 0.0010 *** | 0.0096 *** | 0.0093 *** | −0.0007 ** | 0.0092 *** |
| | (9.5945) | (6.2406) | (9.9844) | (9.5945) | (−2.0318) | (9.5331) |
| STATE | 0.0173 ** | −0.0016 | 0.0167 ** | 0.0173 ** | 0.0210 *** | 0.0212 *** |
| | (2.1599) | (−1.1974) | (2.0914) | (2.1599) | (7.3687) | (2.6541) |
| DUAL | 0.0009 ** | −0.0001 | 0.0009 ** | 0.0009 ** | −0.0003 * | 0.0009 ** |
| | (2.0964) | (−0.7301) | (2.0550) | (2.0964) | (−1.8874) | (1.9703) |
| Top1 | 0.0023 *** | −0.0001 ** | 0.0023 *** | 0.0023 *** | −0.0003 *** | 0.0022 *** |
| | (9.6472) | (−2.1275) | (9.5335) | (9.6472) | (−3.5132) | (9.4217) |
| GDPgrt | 0.2207 * | −0.0357 | 0.2078 | 0.2207 * | −0.0070 | 0.2194 * |
| | (1.6953) | (−1.5981) | (1.5970) | (1.6953) | (−0.1503) | (1.6899) |
| INDUSTRY FE | YES | YES | YES | YES | YES | YES |
| YEAR FE | YES | YES | YES | YES | YES | YES |
| Intercept | −6.4900 *** | −0.0795 *** | −6.5189 *** | −6.4900 *** | 0.2615 *** | −6.4413 *** |
| | (−63.7267) | (−4.6086) | (−64.1299) | (−63.7267) | (7.1625) | (−63.4894) |
| N | 26,245 | 26,245 | 26,245 | 26,245 | 26,245 | 26,245 |
| Adj $R^2$ | 0.7326 | 0.1912 | 0.7337 | 0.7326 | 0.0956 | 0.7338 |
| F | 1205.62 | 69.7400 | 1197.23 | 1205.62 | 34.6514 | 1192.96 |
| Sobel Test | | 0.0015 *** (z = 3.164) | | | 0.0011 ** (z = 2.213) | |
| Goodman Test 1 | | 0.0015 *** (z = 3.15) | | | 0.0011 ** (z = 2.204) | |
| Goodman Test 2 | | 0.0015 *** (z = 3.178) | | | 0.0011 ** (z = 2.222) | |
| Mediating effect | | 7.04% | | | 5.15% | |
| The ratio of indirect to direct effects | | 7.58% | | | 5.43% | |

Note: t-statistics in parentheses. *** $p < 0.01$, ** $p < 0.05$, and * $p < 0.1$.

*5.3. Endogeneity Mitigation and Robustness Checks*

5.3.1. Alternative Measurement for High-Quality Development

In order to ensure the robustness of the empirical results, we replaced the proxy for enterprise high-quality development. More specifically, we used Tobin's Q value of the enterprise (*TobinQ*) as the proxy for high-quality development. *TobinQ* was calculated as the market value divided by total assets. Table 4 shows the robustness test results.

**Table 4.** Alternative measurement for high quality development.

| | (1) TobinQ | (2) EM_Accrual | (3) TobinQ | (4) TobinQ | (5) EM_Real | (6) TobinQ |
|---|---|---|---|---|---|---|
| HSR | 0.0307 ** | −0.0037 *** | 0.0300 ** | 0.0307 ** | −0.0067 *** | 0.0275 * |
| | (2.0538) | (−3.0727) | (2.0073) | (2.0538) | (−2.5891) | (1.8462) |
| EM_Accrual | | | −0.1890 ** | | | |
| | | | (−2.1445) | | | |
| EM_Real | | | | | | −0.4817 *** |
| | | | | | | (−11.3219) |
| SIZE | −0.1689 *** | 0.0034 *** | −0.1682 *** | −0.1689 *** | −0.0241 *** | −0.1805 *** |
| | (−14.9095) | (4.2854) | (−14.8389) | (−14.9095) | (−13.7562) | (−15.9668) |
| LEV | −0.5603 *** | −0.0371 *** | −0.5673 *** | −0.5603 *** | 0.1375 *** | −0.4941 *** |
| | (−10.5565) | (−10.2355) | (−10.6331) | (−10.5565) | (18.2467) | (−9.2775) |
| ROE | −0.0665 | 0.1650 *** | −0.0353 | −0.0665 | −0.2049 *** | −0.1651 *** |
| | (−1.0711) | (33.1355) | (−0.5548) | (−1.0711) | (−21.5499) | (−2.6488) |
| Growth | 0.5693 *** | 0.0352 *** | 0.5760 *** | 0.5693 *** | 0.0412 *** | 0.5892 *** |
| | (18.8639) | (12.6645) | (18.8962) | (18.8639) | (6.8127) | (19.5511) |
| BtoM | −5.2062 *** | 0.0069 ** | −5.2049 *** | −5.2062 *** | 0.2031 *** | −5.1083 *** |
| | (−101.3000) | (2.0539) | (−101.2300) | (−101.3000) | (26.8115) | (−99.0689) |
| Cashflow | −0.0492 *** | 0.0009 *** | −0.0491 *** | −0.0492 *** | −0.0008 *** | −0.0496 *** |
| | (−26.8964) | (5.6353) | (−26.7519) | (−26.8964) | (−2.5769) | (−26.9148) |
| STATE | −0.2440 *** | −0.0015 | −0.2443 *** | −0.2440 *** | 0.0201 *** | −0.2344 *** |
| | (−16.2866) | (−1.1375) | (−16.2978) | (−16.2866) | (7.2662) | (−15.7104) |
| DUAL | −0.0054 *** | −0.0001 | −0.0054 *** | −0.0054 *** | −0.0004 *** | −0.0056 *** |
| | (−6.4283) | (−0.7657) | (−6.4394) | (−6.4283) | (−2.7714) | (−6.6662) |
| Top1 | 0.0039 *** | −0.0001 ** | 0.0038 *** | 0.0039 *** | −0.0003 *** | 0.0037 *** |
| | (8.1860) | (−2.0688) | (8.1500) | (8.1860) | (−3.8780) | (7.8813) |
| GDPgrt | 0.9272 *** | −0.0365 * | 0.9204 *** | 0.9272 *** | −0.0288 | 0.9134 *** |
| | (3.8586) | (−1.6971) | (3.8318) | (3.8586) | (−0.6475) | (3.8064) |
| INDUSTRY FE | YES | YES | YES | YES | YES | YES |
| YEAR FE | YES | YES | YES | YES | YES | YES |
| Intercept | 9.4083 *** | −0.0611 *** | 9.3967 *** | 9.4083 *** | 0.2845 *** | 9.5453 *** |
| | (40.9787) | (−3.6910) | (40.8847) | (40.9787) | (8.1607) | (41.6470) |
| N | 28,062 | 28,062 | 28,062 | 28,062 | 28,062 | 28,062 |
| Adj $R^2$ | 0.6440 | 0.1927 | 0.6441 | 0.6440 | 0.0928 | 0.6465 |
| F | 350.1929 | 73.4922 | 345.3974 | 350.1929 | 34.7401 | 346.6463 |
| Sobel Test | | 0.0007 ** (z = 1.972) | | | 0.0032 ** (z = 2.572) | |
| Goodman Test 1 | | 0.0007 * (z = 1.913) | | | 0.0032 ** (z = 2.565) | |
| Goodman Test 2 | | 0.0007 ** (z = 2.036) | | | 0.0032 *** (z = 2.578) | |
| Mediating effect | | 2.27% | | | 10.43% | |
| The ratio of indirect to direct effects | | 2.32% | | | 11.64% | |

Note: t-statistics in parentheses. *** $p < 0.01$, ** $p < 0.05$, and * $p < 0.1$.

The results in Table 4 showed that the opening of the *HSR* significantly enhanced firms high-quality development, and the information disclosure quality played the mediating role in such a relationship. Thus, our results for H1 and H2 were robust after using an alternative measure of high-quality development.

### 5.3.2. Endogeneity Mitigation

The opening of an *HSR* can be regarded as exogenous in the empirical test, but in fact the opening of an *HSR* station is closely related to regional economy, geographical location, and many other factors. In particular, cities with greater economic strength and more administrative say are more likely to open *HSRs*. We adopted three methods to alleviate the endogeneity problems: exclusion of municipalities and provincial capitals, the propensity score matching (*PSM*) method, and instrumental variable (IV) regressions.

First of all, we excluded municipalities and provincial capitals from our sample and reexamined the impact of *HSR* on the *TFP*. The results are shown in Table 5.

**Table 5.** Endogeneity mitigation: Excluding municipalities and provincial capitals.

| | (1) TFP | (2) EM_Accrual | (3) TFP | (4) TFP | (5) EM_Real | (6) TFP |
|---|---|---|---|---|---|---|
| HSR | 0.0801 *** | −0.0061 *** | 0.0781 *** | 0.0801 *** | −0.0099 ** | 0.0782 *** |
| | (7.3354) | (−3.1835) | (7.1730) | (7.3354) | (−2.5742) | (7.1879) |
| EM_Accrual | | | −0.3231 *** | | | |
| | | | (−5.8704) | | | |
| EM_Real | | | | | | −0.1946 *** |
| | | | | | | (−7.3473) |
| SIZE | 0.6171 *** | 0.0056 *** | 0.6189 *** | 0.6171 *** | −0.0263 *** | 0.6120 *** |
| | (89.0689) | (4.7555) | (89.5819) | (89.0689) | (−10.4845) | (88.2966) |
| LEV | 0.6406 *** | −0.0324 *** | 0.6301 *** | 0.6406 *** | 0.1443 *** | 0.6686 *** |
| | (20.0182) | (−6.3031) | (19.7254) | (20.0182) | (13.9138) | (20.6565) |
| ROE | 0.8832 *** | 0.1664 *** | 0.9369 *** | 0.8832 *** | −0.1823 *** | 0.8477 *** |
| | (19.9895) | (23.9288) | (20.6012) | (19.9895) | (−14.3347) | (19.1393) |
| Growth | −0.1220 *** | 0.0300 *** | −0.1123 *** | −0.1220 *** | 0.0390 *** | −0.1144 *** |
| | (−7.7680) | (7.8867) | (−7.1147) | (−7.7680) | (4.7330) | (−7.2546) |
| BtoM | −0.1074 *** | −0.0038 | −0.1086 *** | −0.1074 *** | 0.2076 *** | −0.0670 ** |
| | (−3.8029) | (−0.8007) | (−3.8515) | (−3.8029) | (20.1341) | (−2.3366) |
| Cashflow | 0.0096 *** | 0.0010 *** | 0.0100 *** | 0.0096 *** | −0.0021 *** | 0.0092 *** |
| | (5.5157) | (3.4632) | (5.7283) | (5.5157) | (−3.5306) | (5.2975) |
| STATE | 0.0068 | −0.0031 | 0.0058 | 0.0068 | 0.0094 ** | 0.0086 |
| | (0.6279) | (−1.6376) | (0.5370) | (0.6279) | (2.5401) | (0.7986) |
| DUAL | 0.0015 *** | −0.0001 | 0.0015 *** | 0.0015 *** | −0.0005 ** | 0.0014 ** |
| | (2.6967) | (−0.8574) | (2.6521) | (2.6967) | (−2.4706) | (2.5267) |
| Top1 | 0.0014 *** | −0.0001 | 0.0014 *** | 0.0014 *** | −0.0005 *** | 0.0013 *** |
| | (4.2885) | (−1.0498) | (4.2399) | (4.2885) | (−4.0902) | (4.0129) |
| GDPgrt | 0.7840 *** | −0.0181 | 0.7782 *** | 0.7840 *** | 0.0401 | 0.7918 *** |
| | (4.3161) | (−0.5754) | (4.2870) | (4.3161) | (0.6519) | (4.3714) |
| INDUSTRY FE | YES | YES | YES | YES | YES | YES |
| YEAR FE | YES | YES | YES | YES | YES | YES |
| Intercept | −6.2981 *** | −0.0989 *** | −6.3301 *** | −6.2981 *** | 0.3440 *** | −6.2312 *** |
| | (−44.1146) | (−4.1199) | (−44.4586) | (−44.1146) | (6.9460) | (−43.7497) |
| N | 13,993 | 13,993 | 13,993 | 13,993 | 13,993 | 13,993 |
| Adj $R^2$ | 0.7308 | 0.2100 | 0.7317 | 0.7308 | 0.1038 | 0.7321 |
| F | 608.8711 | 42.9796 | 602.2675 | 608.8711 | 20.7214 | 601.2643 |
| Sobel Test | | 0.0020 *** (z = 2.901) | | | 0.0019 ** (z = 2.474) | |
| Goodman Test 1 | | 0.0020 *** (z = 2.876) | | | 0.0019 ** (z = 2.457) | |
| Goodman Test 2 | | 0.0020 *** (z = 2.928) | | | 0.0019 ** (z = 2.491) | |
| Mediating effect | | 2.44% | | | 2.40% | |
| The ratio of indirect to direct effects | | 2.51% | | | 2.46% | |

Note: t-statistics in parentheses. *** $p < 0.01$ and ** $p < 0.05$.

Table 5 presents the stepwise regression results after excluding the sample firms that are located in municipalities and provincial capitals to mitigate the endogeneity caused by omitting variables and self-selection bias. The first three columns and the last three columns show the results of the mediating roles of accrual earnings management and real activity earnings management, respectively. We documented that the results in Table 5 were qualitatively similar to the baseline findings and the mediating effect in Tables 2 and 3. The research conclusions were supported.

Secondly, since the regression results may appear to be affected by selective bias owing to self-selection in the *DID* regressions, the *PSM* method was adopted to reduce such endogeneity problems. By matching samples with the same characteristics, we controlled some of the factors that interfered with the *HSR* opening. Table 6 compares the baseline result and the robustness test result when using the *PSM* method; column (1) shows the baseline result, while column (2) presents the *DID* regression results when using the *PSM* sample. The results suggested that the baseline regression results were still robust.

**Table 6.** Endogeneity mitigation: PSM + DID.

| | (1) | (2) |
|---|:---:|:---:|
| | *OLS* | *PSM_OLS* |
| | *TFP* | *TFP* |
| HSR | 0.0214 *** | 0.0224 ** |
| | (2.9283) | (2.2288) |
| SIZE | 0.6351 *** | 0.6376 *** |
| | (129.3674) | (84.5414) |
| LEV | 0.6611 *** | 0.6934 *** |
| | (27.8619) | (19.6628) |
| ROE | 0.9327 *** | 0.9278 *** |
| | (27.3119) | (19.2758) |
| Growth | −0.1175 *** | −0.1111 *** |
| | (−9.5345) | (−6.3717) |
| BtoM | −0.1931 *** | −0.1970 *** |
| | (−9.0936) | (−6.4572) |
| Cashflow | 0.0093 *** | 0.0115 *** |
| | (9.5945) | (7.1550) |
| STATE | 0.0173 ** | 0.0257 ** |
| | (2.1599) | (2.2431) |
| DUAL | 0.0009 ** | 0.0001 |
| | (2.0964) | (0.2270) |
| Top1 | 0.0023 *** | 0.0025 *** |
| | (9.6472) | (7.6400) |
| GDPgrt | 0.2207 * | 0.3128 |
| | (1.6953) | (1.5706) |
| INDUSTRY FE | YES | YES |
| YEAR FE | YES | YES |
| Intercept | −6.4900 *** | −6.5888 *** |
| | (−63.7267) | (−41.3042) |
| N | 26,245 | 20,826 |
| Adj $R^2$ | 0.7326 | 0.7215 |
| F | 1205.62 | 579.9309 |

Note: t-statistics in parentheses. *** $p < 0.01$, ** $p < 0.05$, and * $p < 0.1$.

Thirdly, to further mitigate the endogeneity problems, we chose the geographical slope of prefecture-level cities as the instrumental variable of the *HSR* opening. The reasons for choosing such an instrumental variable were as follows. On the one hand, the geographical slope comprehensively reflects the topographic changes of a certain area, which can indirectly measure the cost of *HSR* construction. The cost of building an *HSR* in plain areas is much lower than that in hilly or mountainous areas, so the terrain slope is an important factor affecting the decision to build an *HSR*. Thus, this index met the correlation requirement. On the other hand, the geographical slope and topographic condition are the natural geographical conditions formed over the long history of a region, which exist objectively and are not directly related to the high-quality development of enterprises. Thus, they satisfied the requirement of exogenesis. Table 7 presents the regression results when using the average slope of prefecture-level cities as the instrumental variable of *HSR* opening. The geographical feature of urban slope (*Slope*) was calculated through ArcGis. The first column shows the first stage regression result of the 2sls regressions, while the second column shows the second stage IV regression result.

**Table 7.** Endogeneity mitigation: IV Method.

| | (1) | (2) |
|---|---|---|
| | **First Stage** HSR | **Second Stage** TFP |
| *Slope* | −0.0052 *** | |
| | (−3.0829) | |
| *HSR* | | 3.1754 *** |
| | | (2.9074) |
| *SIZE* | −0.0112 *** | 0.6704 *** |
| | (−2.7522) | (37.3592) |
| *LEV* | 0.1522 *** | 0.1909 |
| | (8.5725) | (1.0920) |
| *ROE* | 0.0486 ** | 0.7767 *** |
| | (2.5185) | (9.0230) |
| *Growth* | −0.0180 ** | −0.0594 |
| | (−2.0467) | (−1.6367) |
| *BtoM* | 0.0813 *** | −0.4597 *** |
| | (4.6213) | (−4.3093) |
| *Cashflow* | 0.0093 *** | −0.0202 * |
| | (10.2204) | (−1.8838) |
| *STATE* | −0.0872 *** | 0.2919 *** |
| | (−12.9481) | (2.9908) |
| *DUAL* | −0.0006 | 0.0027 * |
| | (−1.5609) | (1.9169) |
| *Top1* | −0.0006 *** | 0.0043 *** |
| | (−3.1442) | (4.4321) |
| *GDPgrt* | 0.7997 *** | −2.1112 ** |
| | (7.6893) | (−2.3672) |
| *INDUSTRY FE* | YES | YES |
| *YEAR FE* | YES | YES |
| Intercept | 0.1249 | −6.8655 *** |
| | (1.5346) | (−22.8105) |
| *N* | 26,040 | 26,040 |
| *F* | 443.9940 | 137.5100 |

Note: t-statistics in parentheses. *** $p < 0.01$, ** $p < 0.05$, and * $p < 0.1$.

Table 7 column (1) shows that there was a significant negative correlation between the opening of the *HSR* and the slope of prefecture-level cities, which was consistent with expectations. The result in column (2) revealed that after using *Slope* as the instrumental variable of *HSR*, *HSR* still had a significant positive effect on *TFP*, and it was significant at the 1% level. The 2sls IV regression results indicated that our baseline results were relatively robust.

### 5.4. Additional Tests

China is made up of 31 provinces, municipalities, and autonomous regions, and the business environment varies greatly from region to region. Enterprises in regions with a better business environment might receive greater government support for their development and are more likely to take advantage of the convenience of *HSR* to create firm value. A better regional business environment can facilitate the operation and development of enterprises, and will magnify the positive effects brought by the opening of *HSR*. Thus, based on the specific institutional background of China, we further examined whether the regional business environment moderated the impact of *HSR* on the high-quality development of enterprises, and we expected that the impact of *HSR* opening on the high-quality development of enterprises would be more significant for enterprises in areas with a better business environment.

Table 8 presents the additional test results based on the moderating effect of the business environment. We divided the full sample into two subsamples of a better business



environment group (above the median score of "The relationship between government and market") and a poorer business environment group (below the median score of "The relationship between government and market") according to the marketization index proposed by Wang et al. [52]. Panel A shows the difference in *HSR's* impact on *TFP* between the two subsamples, while Panel B shows the mediating role of information disclosure quality in the impact of *HSR* on *TFP* in the better business environment subsample.

**Table 8.** Additional test based on business environment.

| **Panel A** | | |
|---|---|---|
| | **(1)** | **(2)** |
| | **Better Business Environment** *TFP* | **Poorer Business Environment** *TFP* |
| *HSR* | 0.0278 *** | 0.0018 |
| | (2.7515) | (0.1638) |
| *SIZE* | 0.6096 *** | 0.6559 *** |
| | (93.0630) | (92.1740) |
| *LEV* | 0.7663 *** | 0.5758 *** |
| | (26.9277) | (18.7423) |
| *ROE* | 0.9472 *** | 0.8671 *** |
| | (27.4646) | (26.7122) |
| *Growth* | −0.1554 *** | −0.0723 *** |
| | (−11.1310) | (−4.9185) |
| *BtoM* | −0.1379 *** | −0.2500 *** |
| | (−4.9308) | (−8.3255) |
| *Cashflow* | 0.0082 *** | 0.0091 *** |
| | (5.4409) | (5.8172) |
| *STATE* | 0.0205 * | 0.0547 *** |
| | (1.8774) | (4.7805) |
| *DUAL* | 0.0014 ** | 0.0005 |
| | (2.3461) | (0.7430) |
| *Top1* | 0.0013 *** | 0.0032 *** |
| | (3.9477) | (9.1449) |
| *GDPgrt* | 0.4672 ** | 0.4557 *** |
| | (2.1183) | (2.7780) |
| *INDUSTRY FE* | YES | YES |
| *YEAR FE* | YES | YES |
| Intercept | −5.9930 *** | −6.9880 *** |
| | (−43.6491) | (−48.8259) |
| *N* | 14,307 | 11,938 |
| Adj $R^2$ | 0.7218 | 0.7531 |
| *F* | 546.8600 | 536.4757 |
| Mean difference test | Chi2(1) = 3.08 Prob > chi2 = 0.0792 | |

| Panel B | | | | | | |
|---|---|---|---|---|---|---|
| | (1) *TFP* | (2) *EM_Accrual* | (3) *TFP* | (4) *TFP* | (5) *EM_Real* | (6) *TFP* |
| *HSR* | 0.0278 *** | −0.0043 ** | 0.0262 ** | 0.0278 *** | −0.0146 *** | 0.0251 ** |
| | (2.6291) | (−2.4728) | (2.4815) | (2.6291) | (−3.9532) | (2.3859) |
| *EM_Accrual* | | | −0.3854 *** | | | |
| | | | (−6.9288) | | | |
| *EM_Real* | | | | | | −0.1851 *** |
| | | | | | | (−7.1554) |
| *SIZE* | 0.6096 *** | 0.0034 *** | 0.6109 *** | 0.6096 *** | −0.0229 *** | 0.6054 *** |
| | (89.8889) | (3.0647) | (90.3719) | (89.8889) | (−9.0838) | (89.5498) |

**Table 8.** *Cont.*

| Panel A | | | | | | |
|---|---|---|---|---|---|---|
| | **(1)** | | | **(2)** | | |
| | **Better Business Environment** | | | **Poorer Business Environment** | | |
| | *TFP* | | | *TFP* | | |
| LEV | 0.7663 *** | −0.0365 *** | 0.7522 *** | 0.7663 *** | 0.1512 *** | 0.7943 *** |
| | (23.2135) | (−6.9759) | (22.8589) | (23.2135) | (13.9968) | (23.8401) |
| ROE | 0.9472 *** | 0.1825 *** | 1.0175 *** | 0.9472 *** | −0.2425 *** | 0.9023 *** |
| | (18.2247) | (23.9052) | (19.1859) | (18.2247) | (−15.5349) | (17.3155) |
| Growth | −0.1554 *** | 0.0349 *** | −0.1420 *** | −0.1554 *** | 0.0497 *** | −0.1462 *** |
| | (−9.6792) | (9.2166) | (−8.7563) | (−9.6792) | (6.1303) | (−9.0384) |
| BtoM | −0.1379 *** | 0.0087 * | −0.1345 *** | −0.1379 *** | 0.2137 *** | −0.0983 *** |
| | (−4.7085) | (1.8108) | (−4.6040) | (−4.7085) | (19.5790) | (−3.3003) |
| Cashflow | 0.0082 *** | 0.0010 *** | 0.0086 *** | 0.0082 *** | −0.0003 | 0.0081 *** |
| | (5.7589) | (4.5268) | (6.0489) | (5.7589) | (−0.5722) | (5.7608) |
| STATE | 0.0205 * | −0.0052 *** | 0.0185 | 0.0205 * | 0.0218 *** | 0.0245 ** |
| | (1.8071) | (−2.6978) | (1.6338) | (1.8071) | (5.3891) | (2.1679) |
| DUAL | 0.0014 ** | 0.0000 | 0.0014 ** | 0.0014 ** | −0.0000 | 0.0014 ** |
| | (2.4004) | (0.2799) | (2.4227) | (2.4004) | (−0.2141) | (2.3885) |
| Top1 | 0.0013 *** | −0.0001 | 0.0012 *** | 0.0013 *** | −0.0001 | 0.0012 *** |
| | (3.9588) | (−1.0911) | (3.8935) | (3.9588) | (−1.2111) | (3.8842) |
| GDPgrt | 0.4672 * | −0.0172 | 0.4606 * | 0.4672 * | 0.0751 | 0.4812 ** |
| | (1.9544) | (−0.4285) | (1.9309) | (1.9544) | (0.8837) | (2.0181) |
| INDUSTRY FE | YES | YES | YES | YES | YES | YES |
| YEAR FE | YES | YES | YES | YES | YES | YES |
| Intercept | −5.9930 *** | −0.0746 *** | −6.0217 *** | −5.9930 *** | 0.1999 *** | −5.9560 *** |
| | (−41.5354) | (−3.1058) | (−41.8396) | (−41.5354) | (3.8463) | (−41.5419) |
| N | 14,307 | 14,307 | 14,307 | 14,307 | 14,307 | 14,307 |
| Adj $R^2$ | 0.7218 | 0.1873 | 0.7230 | 0.7218 | 0.1054 | 0.7230 |
| F | 611.1233 | 39.6044 | 608.4060 | 611.1233 | 20.2517 | 604.7388 |
| Sobel Test | | 0.0017 ** (z = 2.345) | | | 0.0027 *** (z = 3.561) | |
| Goodman Test 1 | | 0.0017 ** (z = 2.329) | | | 0.0027 *** (z = 3.539) | |
| Goodman Test 2 | | 0.0017 ** (z = 2.362) | | | 0.0027 *** (z = 3.584) | |
| Mediating effect | | 5.96% | | | 9.70% | |
| The ratio of indirect to direct effects | | 6.34% | | | 10.75% | |

Note: t-statistics in parentheses. *** $p < 0.01$, ** $p < 0.05$, and * $p < 0.1$.

The results in Table 8 Panel A showed that the coefficient of *HSR* in column (1) was significantly positive, indicating that the positive effect of *HSR* opening on firms' *TFP* was significant for enterprises that were located in better business environments. The coefficient of *HSR* in column (2) was not significant, which means that the impact of *HSR* was negligible for enterprises located in cities with a poorer business environment. The mean difference test result showed that there was significant difference in regression results between the two groups. Panel B shows the successive regression results of the mediating effect of information disclosure quality on the way *HSR* affected *TFP* for the better business environment group. The results were similar to the results in Tables 2 and 3, revealing that information disclosure quality played a partial mediating role in the way the *HSR* opening enhanced enterprises' high-quality development.

## 6. Discussion

From the perspective of economic and business sustainability, transportation infrastructure construction is an important means to promote high-quality economic development in China. In this study, we examined how transportation infrastructure construction influenced the high-quality development of enterprises in the context of an emerging financial market, China. Using the opening of *HSR* as a quasi-natural experiment and

data from Chinese listed enterprises during the period of 2003–2019, we found that the opening of an *HSR* that reduces the travel time and information cost could promote the high-quality development of Chinese listed enterprises. The results of the mediating effect analysis proved that the increased information disclosure quality of enterprises brought by the *HSR* openings was one specific internal channel in such a relationship. The result was robust when we used an alternative measurement for high-quality development of enterprises and proposed a placebo test. To address the endogenous issue, we adopted three methods, including exclusion of municipalities and provincial capitals, the *PSM* method, and IV regressions. Additionally, we also found evidence of the moderating effect of the regional business environment. The results showed that the mediating effect of information disclosure quality on the way *HSR* affects *TFP* was more significant for enterprises located in cities with a better business environment.

This research contributes to the burgeoning literature on the economic and social effects of *HSR*, and extends the current *HSR*-related literature by considering its micro-level economic consequences on the high-quality development of enterprises. This research provides empirical evidence for the effectiveness of transportation infrastructure construction in China. The overall findings indicated that local infrastructure construction is an important factor that cannot be ignored in achieving high-quality development of enterprises, as well as the economic and business sustainability. In addition, from the perspective of the quality of information disclosure, this paper provides insight for corporate managers on how to accelerate the realization of the high-quality development of enterprises. Our research findings will also persuade local government to attach more importance to optimizing business environment reforms and helping local enterprises to achieve high-quality development through a better business environment.

There are some research limitations. First, as for the measurement for high-quality development of enterprises, we adopt the *TFP* as the proxy for the dependent variable in the main test, while using Tobin's Q value as the proxy in the robustness test. Although these two indicators could largely represent high-quality development, they cannot describe all of the characteristics of high-quality development. Further research could explore a more appropriate indicator to make the conclusions more convincing. Second, in this research, we choose Chinese A-share listed enterprises as our research sample. Since the A-share market includes multiple economic sectors, the effect of *HSR* on *TFP* may show differences for enterprises belonging to different sectors. However, we do not discuss it further in our research. Future studies could further distinguish and compare the results for samples in different trading sectors.

## 7. Conclusions

Taking the opening of *HSR* as a quasi-natural experiment, this paper uses Chinese A-share listed companies that publicly traded in SSE and SZSE from 2003 to 2019 as research samples to conduct an empirical research on the impact of transportation infrastructure construction on the high-quality development of Chinese enterprises, and further explores the mediating effect of information disclosure quality in such relationship. The conclusions of this research are as follows:

(1) The opening of *HSR* will help promote the high-quality development of enterprises in cities with *HSR*, which is conducive to the sustainable development of China's economy. As part of the transportation infrastructure, the opening of *HSR* can accelerate the flow of personnel and capital, speed up the dissemination of information in different regions, not only effectively improve the information transparency of enterprises, but also provides a convenient channel for outsiders to monitor corporate behavior. Therefore, the opening of *HSR* will help promote high-quality development of enterprises.

(2) The results of the mechanism analysis shows that the improvement of enterprise information disclosure quality brought by the opening of *HSR* is a specific internal channel for the construction of transportation infrastructure promoting the high-quality development of enterprises. Since the opening of *HSR* helps enterprises to actively disclose internal

financial information, and the quality of information disclosure is one of the important factors to promote the high-quality development of enterprises, thus, the quality of information disclosure plays a certain intermediary role in the impact of *HSR* opening on the high-quality development of enterprises.

(3) Since a better regional business environment can facilitate the operation and development of enterprises, this will magnify the positive effects brought by the opening of *HSR*, and thus the mediating effect of information disclosure quality on the relationship of the *HSR* affecting the *TFP* was more significant for enterprises located in cities with a better business environment.

**Author Contributions:** Conceptualization, T.Z. and X.X.; methodology, T.Z.; software, T.Z.; validation, T.Z., X.X. and Q.D.; formal analysis, T.Z. and X.X.; investigation, T.Z. and X.X.; resources, T.Z. and X.X.; data curation, T.Z. and Q.D.; writing—original draft preparation, T.Z. and X.X.; writing—review and editing, T.Z., X.X. and Q.D.; visualization, T.Z. and Q.D.; supervision, X.X. and Q.D.; project administration, X.X.; funding acquisition, T.Z. and X.X. All authors have read and agreed to the published version of the manuscript.

**Funding:** This research was funded by the National Social Science Foundation of China (grant number: 19BGJ001), National Natural Science Foundation of China (grant number: 72102014), Social Science Department of the Ministry of Education of China (grant number: 20YJC630225), and University of Science and Technology Beijing (grant number: FRF-TP-19-065A1).

**Institutional Review Board Statement:** Not applicable.

**Informed Consent Statement:** Not applicable.

**Data Availability Statement:** Our sample included all Chinese firms listed on the Shenzhen and Shanghai stock exchanges. The financial data comes from the China Stock Market and Accounting Research Database (CSMAR), while the regional business environment data comes from Wang et al. (2019).

**Conflicts of Interest:** The authors declare no conflict of interest.

## Appendix A

**Table A1.** Definitions of the variables.

| | Symbols | Description |
|---|---|---|
| Dependent Variable | *TFP* | The total factor productivity of enterprises calculated through Levinsohn–Petrin (LP) method |
| Independent Variables | *HSR* | If the city where the enterprise is located has high-speed rail in that year, *HSR* equals to 1, otherwise equals to 0 |
| Intermediary Variable | *EM_Accrual* *EM_Real* | Accrual earnings management calculated through Modified Jones Model Real activity earnings management calculated through Roychowdhury (2006)'s method |
| Control Variables | *SIZE* | The natural logarithm of total assets |
| | *LEV* | Total debt/Total assets |
| | *ROE* | Net income/Total equity |
| | *GROWTH* | The growth rate of total asset |
| | *BtoM* | Total assets/Market value |
| | *Cashflow* | The cashflow of the enterprise |
| | *STATE* | State-owned firms, *STATE* = 1; otherwise *STATE* = 0 |
| | *DUAL* | The difference between control right and ownership right of a listed company owned by the actual controller |
| | *Top1* | The shareholding ratio of the largest shareholder |
| | *GDPgrt* | The growth rate of annual GDP for each province |

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
