# Peer review of "Transportation Infrastructure Construction and High-Quality Development of Enterprises: Evidence from the Quasi-Natural Experiment of High-Speed Railway Opening in China"

_sustainability, doi:10.3390/su132313316_

Round 1
Reviewer 1 Report
In terms of content, the article is well written. It would be appropriate to explain the abbreviations SSE, SZSE.
Author Response
Dear Reviewer:
We are thankful for the invaluable comments from the reviewer. Those comments are valuable and very helpful for revising and improving our paper. We have studied the comments carefully and have made revisions, which we hope to meet with approval. The revisions made to the manuscript are marked up using the “Track Changes”.
The main responses to the reviewer’s comments (marked in red) are as follows:
Comment
In terms of content, the article is well written. It would be appropriate to explain the abbreviations SSE, SZSE.
Reply:
Thanks so much for the reviewer’s valuable advice.
As for the abbreviations SSE and SZSE, we gave the full names of SSE and SZSE when they first appeared in the main body of the manuscript (please refer to the second paragraph in page 3):
“Taking Chinese A-share listed companies that publicly traded in the Shanghai Stock Exchange (SSE) and Shenzhen Stock Exchange (SZSE) from 2003 to 2019 as the research sample, we adopted a difference-in-difference (DID) approach to empirically examine the influence of HSR opening on firms’ high-quality development as well as the specific influencing mechanism from the perspective of information disclosure quality.”

Reviewer 2 Report
The paper is devoted to the currently important problem connected with the investigation of the influence of the level of transport infrastructure on companies’ development. However, the manuscript is not so relevant for the field of economic and business aspects of sustainability.
The abstract should contain the results of the paper.
The authors used for the investigation the set of financial information, which describe the A-share listed Chinese companies. However, the authors didn’t justify exactly such a choice and what companies (from what economic sector) were analyzed.
The represented results of hypothesis testing are not easy to understand.
The conclusions have a general character.
It would be better to add the discussion part.
The cited references are not so current (mostly not within the last 5 years)
Author Response
Dear Reviewer:
We are thankful for the invaluable comments from the reviewer. Those comments are all valuable and very helpful for revising and improving our paper. We have studied the comments carefully and have made revisions, which we hope to meet with approval. The revisions made to the manuscript are marked up using the “Track Changes”.
The main responses to the reviewer’s comments (marked in red) are as follows:
Comment 1
The paper is devoted to the currently important problem connected with the investigation of the influence of the level of transport infrastructure on companies’ development. However, the manuscript is not so relevant for the field of economic and business aspects of sustainability.
Reply:
Thanks so much for the reviewer’s valuable advice.
Since high-quality development of the economy is an important guarantee for sustainable economic development, therefore, this paper is very relevant to the field of economic and business aspects of sustainability.
According to the suggestion of the reviewer, we have added relevant statements that this research is conducive to promoting the economic and business sustainable development in the revised manuscript. Some of the modifications are as follows:
“High-quality development of the economy is an important guarantee for economic and business sustainability, and the transportation infrastructure construction is an important channel to achieve high-quality development. ……Overall, this research indicates that local infrastructure construction is an important factor that cannot be ignored to achieve high-quality development of enterprises as well as economic sustainability, and this conclusion will be helpful for corporate managers to enhance information disclosure quality as well as for local governments to attach more importance to optimizing business environments to achieve high-quality development and economic sustainability.” (Please refer to Abstract)
“The stage of "high-quality development" is a new development concept featuring innovative, coordinated, green, open, and shared development, which provides an important guarantee for sustainable development of the economy.” (Please refer to the first paragraph of “1. Introduction”)
“Because of China's vast territory, China is currently facing the problem of unbalanced regional economic development. To achieve economic and business sustainability, the train has become the main form of transportation which support large and frequent flow of talent and materials.” (Please refer to the second paragraph of “2. Institutional Background”)
“Secondly, HSR plays a significant social role in China, helping to promote the sustainable development of China's economy.” (Please refer to the fourth paragraph of “2. Institutional Background”)
“From the perspective of economic and business sustainability, transportation infrastructure construction is an important means to promote high-quality economic development in China.” (Please refer to the first paragraph of “6. Discussion”)
“The overall findings indicated that local infrastructure construction is an important factor that cannot be ignored in achieving high-quality development of enterprises as well as the economic and business sustainability.” (Please refer to the second paragraph of “6. Discussion”)
Comment 2
The abstract should contain the results of the paper.
Reply:
Thanks so much for the reviewer’s suggestion. The abstract had already contain the results of this research in the original manuscript, and we have highlighted it in bright yellow in the revised manuscript. The revised abstract is shown below:
“High-quality development of the economy is an important guarantee for economic and business sustainability, and the transportation infrastructure construction is an important channel to achieve high-quality development. Thus, we take the opening of China's high-speed railway (HSR) as a quasi-natural experiment and use the difference-in-difference model to explore the impact and mechanism of HSR on firms’ high-quality development. By using the total factor productivity of enterprises as the proxy for high-quality development, the empirical results show that: (1) The opening of the HSR can significantly promote the high-quality development of enterprises; (2) Information disclosure quality plays a mediating role in such relationship; (3) The impact of HSR on enterprises’ high-quality development is more significant for enterprises located in cities with better business environments. Overall, this research indicates that local infrastructure construction is an important factor that cannot be ignored to achieve high-quality development of enterprises as well as economic sustainability, and this conclusion will be helpful for corporate managers to enhance information disclosure quality as well as for local governments to attach more importance to optimizing business environments to achieve high-quality development and economic sustainability.”
Comment 3
The authors used for the investigation the set of financial information, which describe the A-share listed Chinese companies. However, the authors didn’t justify exactly such a choice and what companies (from what economic sector) were analyzed.
Reply:
Thanks so much for the reviewer’s valuable comment.
For the sample selection, we take all the Chinese A-share listed companies as the initial sample, and then exclude firms in the financial sector such as commercial banks and insurance institutions, financially distressed firms or samples with incomplete data.
The original manuscript did not elaborate on sample selection, and we apologize for the inconvenience caused to the reviewer. We elaborate in Section “4.1 Data and Sample” in the revised manuscript. The modifications are as follows:
“Our research sample contained all the A-share listed firms that publicly traded in the SSE and SZSE during 2003–2019. The sample period began in 2003 because some variables in the regression model were not available prior to 2003. All the financial data were extracted from the China Stock Market and Accounting Research (CSMAR) data-base. To enhance measurement validity, we eliminated the initial sample as follows: (1) firms operating in the financial sector; (2) financially distressed firms; (3) samples with incomplete data. Finally, we obtained 26,245 unbalanced firm-year observations.”
In addition, the sample of this research includes all A-share listed companies except firms in financial sector. We did not divide the samples according to economic sector, nor did we compare the differences between enterprises in different sectors affected by HSR. In section “6. Discussion”, we add a paragraph to explain the limitations of this research, which explains we only use the full sample to conduct analysis. We propose that future studies can further compare the impact of the HSR opening on the high-quality development of enterprises in different economic sectors.
The modifications are as follows:
“There are some research limitations: ……Second, in this research, we choose Chinese A-share listed enterprises as our research sample. Since the A-share market includes multiple economic sectors, the effect of HSR on TFP may show differences for enterprises belonging to different sectors. However, we do not discuss it further in our research. Future studies could further distinguish and compare the results for samples in different trading sectors.”
Comment 4
The represented results of hypothesis testing are not easy to understand.
Reply:
We are thankful for the invaluable comment from the reviewer. For hypothesis 1, we explained the regression results in more detail in the revised manuscript to prove that the empirical regression results support the research hypothesis of this study.
The modifications are as follows:
“The results revealed that the coefficients of HSR in both columns were positive and significant at the 1% level, indicating that the TFP of enterprises in cities with HSR is significantly improved after HSR opening, thus HSR opening promotes the high-quality development of enterprises.” (Please refer to the first paragraph of “5.2.1. HSR and TFP”)
While for hypothesis 2, we adopt a stepwise regression method to examine the mediating role of information disclosure quality. We explained the results in more detail in the revised manuscript as follows:
“Table 3 presents the empirical results for how information disclosure quality mediated the impact of HSR on the high-quality development of enterprises. The first three columns show the mediating effect of information disclosure quality that proxied by accrual earnings management (EM_Accrual), while the last three columns show the mediating effect of information disclosure quality that proxied by real activity earnings management (EM_Real). The coefficient of HSR in column (1) represents that HSR opening has a significant positive influence on TFP; column (2) shows that the opening of HSR significantly inhibit firms’ earnings management, which leading to a better in-formation disclosure quality; and column (3) presents the joint effect of HSR and in-formation disclosure quality on TFP, indicating that HSR and information quality significantly enhanced the TFP of listed enterprises jointly. According to Baron and Kenney [51], the core explanatory variables in the stepwise regressions were all significant, suggesting that information disclosure quality played a mediating role in the way HSR facilitates firms’ TFP, and the mediating effect was approximately 7.04%. The last three columns of Table 3 show the results of the mediating effect of information disclosure quality that were proxied by real activity earnings management. The results in columns (4) to (6) are similar to those in columns (1) to (3), with a mediating effect of 5.15%. Additionally, the Sobel test, Goodman test 1, and Goodman test 2 on the two groups of successive regressions were statistically significant. Therefore, we can conclude that information disclosure quality does play a partial mediating role in the channels through which HSR improves enterprises’ development quality, which supports our H2.” (Please refer to “5.2.2. The Mediating Role of Information Disclosure Quality”)
Comment 5
The conclusions have a general character. It would be better to add the discussion part.
Reply:
We highly appreciate the reviewer’s comment. Following the reviewer’s suggestion, we add section “6. Discussion” and rewrite section “7. Conclusions”, which are shown below:
“6. Discussion
From the perspective of economic and business sustainability, transportation infrastructure construction is an important means to promote high-quality economic development in China. In this study, we examined how transportation infrastructure construction influenced the high-quality development of enterprises in the context of an emerging financial market, China. Using the opening of HSR as a quasi-natural experiment and data from Chinese listed enterprises during the period of 2003–2019, we found that the opening of an HSR that reduces the travel time and information cost could promote the high-quality development of Chinese listed enterprises. The results of the mediating effect analysis proved that the increased information disclosure quality of enterprises brought by the HSR openings was one specific internal channel in such a relationship. The result was robust when we used an alternative measurement for high-quality development of enterprises and proposed a placebo test. To address the endogenous issue, we adopted three methods, including exclusion of municipalities and provincial capitals, the PSM method, and IV regressions. Additionally, we also found evidence of the moderating effect of the regional business environment. The results showed that the mediating effect of information disclosure quality on the way HSR affects TFP was more significant for enterprises located in cities with a better business environment.
This research contributes to the burgeoning literature on the economic and social effects of HSR, and extends the current HSR-related literature by considering its micro-level economic consequences on the high-quality development of enterprises. This research provides empirical evidence for the effectiveness of transportation infra-structure construction in China. The overall findings indicated that local infrastructure construction is an important factor that cannot be ignored in achieving high-quality development of enterprises as well as the economic and business sustainability. In addition, from the perspective of the quality of information disclosure, this paper provides insight for corporate managers on how to accelerate the realization of high-quality development of enterprises. Our research findings will also persuade local government to attach more importance to optimizing business environment reform and help local enterprises achieve high-quality development through a better business environment.
There are some research limitations: First, as for the measurement for high-quality development of enterprises, we adopt TFP as the proxy for the dependent variable in the main test, while using Tobin’s Q value as the proxy in the robustness test. Although these two indicators could largely represent high-quality development, they cannot describe all the characteristics of high-quality development. Further research could explore a more appropriate indicator to make the conclusions more convincing. Second, in this research, we choose Chinese A-share listed enterprises as our research sample. Since the A-share market includes multiple economic sectors, the effect of HSR on TFP may show differences for enterprises belonging to different sectors. However, we do not discuss it further in our research. Future studies could further distinguish and compare the results for samples in different trading sectors.
- Conclusions
Taking the opening of HSR as a quasi-natural experiment, this paper uses Chinese A-share listed companies that publicly traded in SSE and SZSE from 2003 to 2019 as research samples to conduct an empirical research on the impact of transportation infrastructure construction on the high-quality development of Chinese enterprises, and further explores the mediating effect of information disclosure quality in such relationship. The conclusions of this research are as follows:
(1) The opening of HSR will help promote the high-quality development of enterprises in cities with HSR, which is conducive to the sustainable development of China’s economy. As part of the transportation infrastructure, the opening of HSR can accelerate the flow of personnel and capital, speed up the dissemination of information in different regions, not only effectively improve the information transparency of enterprises, but also provides a convenient channel for outsiders to monitor corporate behavior. Therefore, the opening of HSR will help promote high-quality development of enterprises.
(2) The results of the mechanism analysis shows that the improvement of enterprise information disclosure quality brought by the opening of HSR is a specific internal channel for the construction of transportation infrastructure promoting the high-quality development of enterprises. Since the opening of HSR helps enterprises to actively disclose internal financial information, and the quality of information dis-closure is one of the important factors to promote the high-quality development of enterprises, thus, the quality of information disclosure plays a certain intermediary role in the impact of HSR opening on the high-quality development of enterprises.
(3) Since a better regional business environment can facilitate the operation and development of enterprises, and will magnify the positive effects brought by the opening of HSR, thus the mediating effect of information disclosure quality on the relationship HSR affecting TFP was more significant for enterprises located in cities with a better business environment.”
Comment 6
The cited references are not so current (mostly not within the last 5 years)
Reply:
We are thankful for the reviewer’s comment. We have updated the references in the revised manuscript. After modification, the literature in the last five years (2017-2021) accounted for 30/52 of the total references.

Reviewer 3 Report
I read the paper “Transportation infrastructure construction and high-quality development of enterprises: Evidence from the quasi-natural experiment of high-speed railway opening in China” with interest. It is about a relevant topic, well written, clear in the proposed methodology and the findings.
I have only some minor points that I would like to highlight:
1) Section 4.2.1: I cannot understand how TFP can be a proxy of high-quality development. Can authors explain this aspect better?
2) In interpreting results of Table 1, authors claim that with a mean value of TFP equal to 8.07, "the total factor productivity of listed companies in China still has great room for improvements". Can we have a benchmark or a range of values to allow a more appropriate interpretation based on quantitative values?
3) I would suggest authors mentioning also aspects related to equity deriving from the introduction of HSR. There is a huge literature on this topic (see references below) and I think that this is relevant presenting it when dealing with labour market and infrastructures.
References:
Chen, Z., Haynes, K.E., 2017. Impact of high-speed rail on regional economic disparity in China. Journal of Transport Geography 65, 80-91.
Cavallaro, F., Bruzzone, F., Nocera, S., 2020. Spatial and social equity implications for High-Speed Railway lines in Northern Italy. Transportation Research Part A: Policy and Practice, 135, 327-340
Kim, H., Sultana S., 2015. The impacts of high-speed rail extensions on accessibility and spatial equity changes in South Korea from 2004 to 2018. Journal of Transport Geography 45, 48–61.
Author Response
Dear Reviewer:
We are thankful for the invaluable comments from the reviewer. Those comments are all valuable and very helpful for revising and improving our paper. We have studied the comments carefully and have made revisions, which we hope to meet with approval. The revisions made to the manuscript are marked up using the “Track Changes”.
The main responses to the reviewer’s comments (marked in red) are as follows:
Comment 1
Section 4.2.1: I cannot understand how TFP can be a proxy of high-quality development. Can authors explain this aspect better?
Reply:
We are thankful for the reviewer’s comment. According to the suggestion of reviewer, we have elaborated the reasons for using TFP as the proxy for high-quality development of enterprises in section 4.2.1 of the revised manuscript.
The modifications are as follows:
“High-quality development of enterprises refers to the pursuit of a high level and high efficiency of economic value and social value creation, with innovation serves as the first impetus [43]. According to this definition, the total factor productivity (TFP) of enterprises was a suitable proxy for high-quality development, since Robert Merton Solow, a Nobel Prize winner of economics, attributed TFP to technological progress.”
Comment 2
In interpreting results of Table 1, authors claim that with a mean value of TFP equal to 8.07, "the total factor productivity of listed companies in China still has great room for improvements". Can we have a benchmark or a range of values to allow a more appropriate interpretation based on quantitative values?
Reply:
We are thankful for the invaluable comment from the reviewer.
Since there is no definite benchmark for TFP, we concluded that there is still a great space for improvement of TFP level in China according to the TFP values in prior literature. According to the suggestion of the reviewer, to ensure the stringency of the paper, we have deleted this statement in the revised manuscript.
The modifications are as follows:
“The results of Table 1 revealed that the mean TFP was 8.07, and the standard deviation was 1.01. In our sample, 54% of the observations had experienced the opening of HSR. The standard deviations of the control variables Cashflow, DUAL, and Top1 were relatively large, which indicated that the cash flow and the corporate governance of each sample firm were quite different.” (Please refer to section “5.1 Summary Statistics”)
Comment 3
I would suggest authors mentioning also aspects related to equity deriving from the introduction of HSR. There is a huge literature on this topic (see references below) and I think that this is relevant presenting it when dealing with labour market and infrastructures.
References:
Chen, Z., Haynes, K.E., 2017. Impact of high-speed rail on regional economic disparity in China. Journal of Transport Geography 65, 80-91.
Cavallaro, F., Bruzzone, F., Nocera, S., 2020. Spatial and social equity implications for High-Speed Railway lines in Northern Italy. Transportation Research Part A: Policy and Practice, 135, 327-340
Kim, H., Sultana S., 2015. The impacts of high-speed rail extensions on accessibility and spatial equity changes in South Korea from 2004 to 2018. Journal of Transport Geography 45, 48–61.
Reply:
We are thankful for the invaluable comment from the reviewer.
According to the reviewer’s suggestion, in the revised manuscript, we have added literature related to spatial and social equity that brought by HSR.
The modifications are as follows:
“High-Speed Railway (HSR) is a large-scale transportation infrastructure investment that was introduced in China in 2008 to facilitate the flow of information, capital, and labor among cities [3-4], to increase spatial and social equity [5-7], as well as to stimulate economic growth [8-9].” (Please refer to section “1. Introduction”)
Above mentioned references:
- Ke, X.; Chen, H.; Hong, Y.; Hsiao, C. Do China's high-speed-rail projects promote local economy?—New evidence from a panel data approach. China Econ. Rev. 2017, 44, 203–226. https://doi.org/10.1016/j.chieco.2017.02.008
- Lin, Y. Travel costs and urban specialization patterns: evidence from China's high speed railway system. Urban Econ. 2017, 98, 98–123. https://doi.org/10.1016/j.jue.2016.11.002
- Cavallaro, F.; Bruzzone, F.; Nocera, S. Spatial and social equity implications for High-Speed Railway lines in Northern Italy. Transport Res. A-Pol. 2020, 135, 327-340. https://doi.org/10.1016/j.tra.2020.03.028
- Chen, Z.; Haynes, K. E. Impact of high-speed rail on regional economic disparity in China. Transp. Geogr. 2017, 65, 80-91. http://dx.doi.org/10.1016/j.jtrangeo.2017.08.003
- Kim, H.; Sultana, S. The impacts of high-speed rail extensions on accessibility and spatial equity changes in South Korea from 2004 to 2018. Transp. Geogr. 2015, 45, 48–61. http://doi.org/10.1016/j.jtrangeo.2015.04.007
- Qin, Y. ‘No county left behind?’ The distributional impact of high-speed rail upgrades in China. Econ. Geogr. 2017, 17 (3), 489–520. https://doi.org/10.1093/jeg/lbw013
- Wang, Y.; Liang, S.; Kong, D.; Wang, Q. High-speed rail, small city, and cost of debt: firm-level evidence. Pac-Basin. Financ. J. 2019, 57, 101194. http://doi.org/10.1016/j.pacfin.2019.101194
- Yang, X.; Zhang, Z.; Luo, W.; Tang, Z.; Gao, X.; Wan, Z. and Zhang, X. The impact of government role on high-quality innovation development in mainland China. Sustainability 2019, 11, 5780. https://doi.org/10.3390/su11205780
